# Comprehensive Targeted Metabolomic Study in the Lung, Plasma, and Urine of PPE/LPS-Induced COPD Mice Model

**DOI:** 10.3390/ijms23052748

**Published:** 2022-03-02

**Authors:** Hyeon-Young Kim, Hyeon-Seong Lee, In-Hyeon Kim, Youngbae Kim, Moongi Ji, Songjin Oh, Doo-Young Kim, Wonjae Lee, Sung-Hwan Kim, Man-Jeong Paik

**Affiliations:** 1Jeonbuk Branch Institute, Korea Institute of Toxicology, Jeongeup 56212, Korea; hyeonyoung.kim@kitox.re.kr (H.-Y.K.); inhyeon.kim@kitox.re.kr (I.-H.K.); 2College of Veterinary Medicine, Chonnam National University, Gwangju 61186, Korea; 3College of Pharmacy, Chosun University, Gwangju 61452, Korea; dlrudgks1022@kist.re.kr (H.-S.L.); wlee@chosun.ac.kr (W.L.); 4Korea Institute of Science and Technology, Gangneung Institute of Natural Products, Gangneung 25451, Korea; 5College of Pharmacy, Sunchon National University, Suncheon 57922, Korea; unkr2003@naver.com (Y.K.); wlansrl@naver.com (M.J.); osj7797@naver.com (S.O.); 20110074@hdpharm.co.kr (D.-Y.K.); 6Hyundai Pharm, New Drug Discovery Lab, Yongin 17089, Korea

**Keywords:** chronic obstructive pulmonary disease, metabolomics, PPE/LPS-induced COPD exacerbation mice model, receiver operating characteristic analysis, potential biomarker

## Abstract

(1) Background: Progression of chronic obstructive pulmonary disease (COPD) leads to irreversible lung damage and inflammatory responses; however, biomarker discovery for monitoring of COPD progression remains challenging. (2) Methods: This study evaluated the metabolic mechanisms and potential biomarkers of COPD through the integrated analysis and receiver operating characteristic (ROC) analysis of metabolic changes in lung, plasma, and urine, and changes in morphological characteristics and pulmonary function in a model of PPE/LPS-induced COPD exacerbation. (3) Results: Metabolic changes in the lungs were evaluated as metabolic reprogramming to counteract the changes caused by the onset of COPD. In plasma, several combinations of phenylalanine, 3-methylhistidine, and polyunsaturated fatty acids have been proposed as potential biomarkers; the α-aminobutyric acid/histidine ratio has also been reported, which is a novel candidate biomarker for COPD. In urine, a combination of succinic acid, isocitric acid, and pyruvic acid has been proposed as a potential biomarker. (4) Conclusions: This study proposed potential biomarkers in plasma and urine that reflect altered lung metabolism in COPD, concurrently with the evaluation of the COPD exacerbation model induced by PPE plus LPS administration. Therefore, understanding these integrative mechanisms provides new insights into the diagnosis, treatment, and severity assessment of COPD.

## 1. Introduction

Chronic obstructive pulmonary disease (COPD) is a slowly developing respiratory disease that is accompanied by irreversible emphysema with chronic airway inflammation, mucus hypersecretion [1], and airway remodeling, and is one of the growing causes of death [2]. The main causes of COPD include cigarette smoke (CS), air pollution, fine dust, and exposure to various inducers in the workplace environment [3]. The prevalence and global incidence of COPD is approximately 174 million people [4]. Moreover, the annual death from COPD in the United States is more than 120,000 people and about 3.2 million people globally [4,5]. Mortality rates have been increasing significantly [6] due to acute exacerbation of COPD; the main cause is acute reactions caused by bacterial infections [7]. However, due to the lack of understanding of the molecular mechanisms of COPD, an adequate treatment method does not exist [8]. Studies are currently being conducted on various animal models to understand the biochemical mechanism of COPD [9]. The common methods of COPD induction in mice include exposure to CS or a combination inducer of elastase and lipopolysaccharide (LPS). The CS mouse model has the advantage of being able to best reflect the clinical symptoms of COPD, although the method for animal exposure has not yet been standardized. The model that uses the combination inducer, elastase and LPS, enables easy induction of the disease, while also allowing control over the severity of the disease based on the dose of the inducer, but has the disadvantage of reflecting several, but not all, pathophysiological changes, depending on emphysema [9]. The design and selection of an appropriate animal model is critical and should be done based on the purpose of the study.

Among all omics studies, the phenotypic information of a disease is best reflected in metabolomics [10]. Metabolomic research is gaining attention as a tool for improving the biochemical understanding of complex diseases, such as respiratory diseases [8,11]. In metabolomics, profile analysis is generally performed using GC and LC-based mass spectrometry or NMR [12]. It is important to select an analytical instrument based on the properties of the metabolite. In general, LC-based analysis is preferred for non-volatile and polar metabolites [12]. For some metabolites, such as fatty acids and organic acids, GC-based analysis after derivatization is preferred because it is generally more sensitive and selective with the improved MS property than those of non-derivatization [13]. Many metabolomics-based COPD studies have been performed in serum, plasma, lung, and exhaled breath condensate, but the biochemical mechanisms of COPD are not yet fully understood [8]. In this study, the combination inducer method of elastase and LPS was used to evaluate and understand the acute response to COPD. To identify changes in response to COPD with repeated exposure to acute exacerbations, porcine pancreatic elastase (PPE) and LPS were administered once a week or twice for two weeks, respectively. Immunological and histopathological tests were performed to validate the animal model, and metabolite profile analysis based on mass spectrometry (MS) was performed in the lung, plasma, and urine to understand the metabolic mechanisms.

## 2. Results

### 2.1. Lung Weight

As shown in Appendix A, there were significant increases in absolute and relative lung weights in the COPD1 group when compared with those in the VC1 group. In addition, there was a significant increase in the absolute and relative lung weights in the COPD2 group when compared with those in the VC2 group. However, there was no significant difference in lung weights between COPD1 and COPD2 groups.

### 2.2. Lung Function Measurement

The VC2 and COPD2 groups showed a significant increase in functional residual capacity (FRC), total lung capacity (TLC), forced vital capacity (FVC) (Appendix A), and a decrease in the FEV (forced expiratory volume) 100/FVC (Appendix A), when compared with the VC1 and COPD1 groups. TLC and FVC were significantly increased in COPD2 compared with those in the COPD1 group (Appendix A).

### 2.3. Cell Composition and Number in Bronchoalveolar Lavage Fluid (BALF)

The number of total cells, macrophages, neutrophils, and lymphocytes was significantly higher in the COPD1 than in the VC1 group (Appendix A). The same was true between the COPD2 and VC2 groups (Appendix A). There was no significant change in eosinophil count in the COPD1 group neutrophils and eosinophils in the COPD2 group when compared with the respective vehicle control groups (Appendix A). Moreover, there was no significant difference between the COPD1 and COPD2 groups in cell composition and the number in BALF.

### 2.4. Histological Examination

H&E staining showed emphysema and interstitial inflammatory cell infiltration in both COPD1 and COPD2 groups (Appendix A). These histological changes were reflected in an increase in the mean linear intercept (MLI, Appendix A). The MLI was significantly increased in COPD1 and COPD2 groups, compared with the respective vehicle control groups (Appendix A). In both H&E and MLI results, COPD2 demonstrated higher histological lung injury than COPD1.

### 2.5. Metabolomic Alteration in the Lung of PEE/LPS-Induced COPD Exacerbation Model

A total of 69 metabolites were identified in the lungs of the PPE/LPS-induced COPD exacerbation model, and the representative chromatograms are shown in Figure 1A,B. The levels of metabolites were normalized according to the body weight of each mouse. The four groups were completely clustered into two large classes using principal component analysis (PCA) (Appendix A), partial least squares-discriminant analysis (PLS-DA) (Appendix A), and hierarchical clustering heatmaps (Appendix A). The two large classes were clustered according to the control and COPD-induced groups, and subclasses VC1 (blue) and VC2 (sky blue), and COPD1 (red) and COPD2 (pink) were evaluated as similar groups. The levels of the analyzed metabolites were highest mostly in the COPD1 group (Appendix A). Specifically, 49 metabolites (FDR < 0.05), including 16 metabolites (Log2FC > 1), were significantly different between the VC1 and COPD1 groups. Of these, 37 were significantly increased and 12 were significantly decreased in the COPD1 group when compared with the VC1 group (Appendix A). PCA and PLS-DA were performed to evaluate the discrimination between groups, and the clustering of VC1 and COPD1 groups was evident for the 69 metabolites (Figure 2c,d); the top-ranked 15 metabolites obtained by the variable importance in projection (VIP) score of PLS-DA are shown in Figure 2e. A hierarchical clustering heatmap of 69 metabolites showed a clear classification between VC and COPD1. The levels of 29 metabolites (top-29 metabolites of the heatmap in Figure 2a) were largely increased in the COPD1 group compared with the VC1 group and had an important role in clustering the two groups. Quantitative metabolite set enrichment analysis (MSEA) was performed to identify biologically meaningful patterns and was evaluated based on the 49 metabolites (FDR < 0.05). The prognostic power of core metabolites was validated in 40 metabolite sets, including “tyrosine metabolism”,” β-alanine metabolism” and “pyrimidine metabolism” (Appendix A). The results of the quantitative MSEA in the COPD1 group are shown in Appendix A. In the comparison between the VC2 and COPD2 groups, 42 metabolites (FDR < 0.05), including 3 metabolites (Log2FC > 1), were significantly different. The levels of 30 metabolites were significantly increased; 12 metabolites were significantly decreased in the COPD2 group compared to the VC2 group (Appendix A). In PCA and PLS-DA for 69 metabolites, VC2 and COPD2 groups were clearly classified (Figure 2f,g); the top-ranked 15 metabolites are shown in Figure 2h. In a hierarchical clustering heatmap performed with 69 metabolites, clear clustering was shown based on the increased levels of 26 metabolites (Figure 2b). In quantitative MSEA for the 42 metabolites (FDR < 0.05) in COPD2, the prognostic power of core metabolites was validated in 40 metabolite sets, including “pyrimidine metabolism”, tyrosine metabolism”, “butanoate metabolism”, and phenylalanine metabolism” (Appendix A). The results of the quantitative MSEA in the COPD2 group are shown in Appendix A.

### 2.6. Metabolomic Alteration in Plasma of PEE/LPS-Induced COPD Exacerbation Model

A total of 68 metabolites normalized according to weight were identified in the plasma of the PPE/LPS-induced COPD exacerbation model, and the representative chromatograms are shown in Figure 1A,B. In the PCA, PLS-DA, and hierarchical clustering heatmaps, the four groups were clustered into two groups (Appendix A). In plasma, unlike in lung tissue, clustering was classified based on the week of the study regardless of COPD induction, similar to VC1 and COPD1, and VC2 and COPD2 (Appendix A). The levels of the metabolites investigated were mostly highest in the VC1 and COPD1 groups (Appendix A). The levels of seven metabolites (3-methylhistidine, α-aminobutyric acid, γ-linolenic acid, glutathione, GSSG, histidine, and malonic acid) in the VC1 and COPD1 groups and the levels of eight metabolites (4-hydroxyphenylaectic acid, alanine, erucic acid, glycine, nervonic acid, octadecanoic acid, pyruvic acid, and serine) in the VC2 and COPD2 groups were significantly different based on the Wilcoxon rank-sum test (*p* < 0.05; Appendix A). PCA did not clearly separate the two groups (Figure 3a,g), whereas PLS-DA showed a relatively clear separation within the VC1 and COPD1, and VC2 and COPD2 groups, respectively (Figure 3d,j). The 15 top-ranked metabolites by PLS-DA are shown in Figure 3c,i, respectively. The nine top-ranked metabolites based on the VIP score of PLS-DA were selected for the hierarchical clustering heatmap; the VC1 and COPD1, and VC2 and COPD2 groups were fairly distinctly clustered, respectively (Figure 3b,h).

In the univariate ROC curve analysis of 68 metabolites and their 20 metabolite ratios, metabolite combinations with large changes in Log2FC and AUC > 0.7, and significant *p*-values were given priority and evaluated. Different metabolite combinations were used for COPD1 and COPD2 groups. In the COPD1 group, 40 biomarkers, including malonic acid/3-methylhistidine (AUC = 0.99, Log2FC = −5.38), fumaric acid/docosapentaenoic acid (AUC = 0.93, Log2FC = −4.96), glutamic acid/histidine (AUC = 0.96, Log2FC = −4.47), malonic acid/erucic acid (AUC = 0.92, Log2FC = −1.55), and 4-hydroxyphenyllactic acid/3-methylhistidine (AUC = 0.90, Log2FC = −1.31) were used and shown in Appendix A. In the COPD2 group, 43 biomarkers, including 4-hydroxyphenylacetic acid (AUC = 0.88, Log2FC = 2.07), tyrosine/alanine (AUC = 0.99, Log2FC = 1.49), valine/alanine (AUC = 0.95, Log2FC = 0.74), phenylalanine (AUC = 0.99, Log2FC = 0.66), and nervonic acid (AUC = 0.79, Log2FC = 0.57) were used and shown in Appendix A. Multivariate ROC curve analysis (Explorer) was performed with 68 metabolites and their metabolite ratios. Multivariate ROC curve analysis may provide better predictive biomarker model creation and evaluation than univariate ROC curve analysis. The multiple biomarkers combined top-3 biomarkers (i.e., malonic acid/3-methylhistidine, malonic acid/docosapentaenoic acid, and malonic acid/oxaloacetic acid) for distinguishing VC1 and COPD1 groups could achieve AUC = 0.973 (Figure 3e,f). The multiple biomarkers combined top-three biomarkers (i.e., phenylalanine/α-aminobutyric acid, 4-hydroxyphenylaceic acid/α-aminobutyric acid, and phenylalanine/glutamine) for discriminating VC2 and COPD2 groups could achieve AUC = 0.945 (Figure 3k,l).

### 2.7. Metabolomic Alteration in the Urine of PEE/LPS-Induced COPD Exacerbation Model

A total of 60 metabolites were identified in the urine of the PPE/LPS-induced COPD exacerbation model, and the representative chromatograms are shown in Figure 1A,B. The levels of metabolites were normalized according to the creatinine level (1 ng) of each mouse. The levels of 16 metabolites (including arachidonic acid, malic acid, and fumaric acid) in comparison with the VC1 and COPD1 groups, and the levels of 26 metabolites (including cystine, oxaloacetic acid, and proline) compared in VC2 and COPD2 were significantly different in Wilcoxon rank-sum test (*p* < 0.05; Appendix A). The four groups were not clustered using PCA, PLS-DA, or hierarchical clustering heatmaps (Appendix A).

Univariate ROC curve analysis was performed for 60 metabolites and their 20 metabolite ratios. In the COPD1 group, 44 biomarkers, including glycolic acid/malic acid (AUC = 0.93, Log2FC = −4.25), malic acid (AUC = 0.90, Log2FC = 2.01), fumaric acid/oxaloacetic acid (AUC = 0.93, Log2FC = 1.17), malic acid/2-hydroxyglutaric acid (AUC = 0.96, Log2FC = 0.91), and fumaric acid/4-hydroxyphenyllactic acid (AUC = 0.91, Log2FC = 0.72) were used and are shown in Appendix A. In the COPD2 group, 63 biomarkers, including tryptophan/proline (AUC = 0.91, Log2FC = 5.17), leucine/β-aminobutyric acid (AUC = 0.92, Log2FC = −4.83), β-alanine/histidine (AUC = 0.90, Log2FC = 3.71), threonine/β-alanine (AUC = 0.93, Log2FC = −2.49), and valine/β-aminobutyric acid (AUC = 0.93, Log2FC = −0.83) were used and shown in Appendix A. Multivariate ROC curve analysis (Explorer) was performed with 60 metabolites and their 20 metabolite ratios. The multiple biomarkers combined top-3 biomarkers (i.e., succinic acid/malic acid, malic acid/cis-aconitic acid, and citrulline/targinine) for discriminating VC1 and COPD1 groups could achieve AUC = 0.949 (Figure 4a,b). The multiple biomarkers combined top-3 biomarkers (i.e., lactic acid/cystine, glycine/β-aminobutyric acid, and valine/β-aminobutyric acid) for distinguishing VC2 and COPD2 groups could achieve AUC = 0.907 (Figure 4c,d).

## 3. Discussion

### 3.1. PEE/LPS-Induced COPD Exacerbation Model

Exacerbations are a major characteristic of COPD patients that, if repeated, can cause a rapid decline in lung function and even lead to death [14]. As there is a clear need to understand the underlying mechanisms driving exacerbations, we tested a model of COPD exacerbation that resembles COPD patients with pulmonary and extrapulmonary compromise. CS is a major causative factor in COPD pathogenesis. However, CS-induced animal models are difficult to control, require long-term exposure [15]. The most common putative causes of acute exacerbations in COPD patients are endotoxins formed by gram-negative bacteria [16]. LPS constitute an important part of the outer membrane of gram-negative bacteria [17]; recent studies have reported that intratracheal instillation of LPS can cause airway inflammation and lung damage [18]. According to the study by Based on previous reports, in this experimental mouse model, COPD was induced by intratracheal instillation of PPE, and exacerbation was induced by intratracheal instillation of LPS [19,20]. Neutrophil accumulation and decline in pulmonary function are typical features of exacerbations in human COPD [21]; our study has been similarly evaluated with the results of the previous COPD animal model. Thus, our mouse COPD model reflects the major pathological features of COPD exacerbation in humans. Although several animal models have been established to simulate the injury response of COPD, no in vivo model can exactly mimic human COPD exacerbation. The strength of this study is that it is the first to evaluate not only morphological characteristics and pulmonary function, but also to investigate the metabolic changes in a model of PPE/LPS-induced COPD exacerbation. In addition, severe exacerbation of COPD patients required long term exposure to CS, air pollution, fine dust, and many chemicals, whereas our COPD mouse model could, to some extent, reflect the pathology and physiology of human COPD in a short modeling time.

### 3.2. Metabolic Changes in Lung

In this study, target metabolomics analysis was performed on 44 organic metabolites, including amino acids using LC-MS/MS and 18 organic acids and 24 fatty acids using GC-MS/MS. A total of 86 metabolites can be evaluated importantly for their relevance to energy metabolism based on the TCA cycle; metabolites previously evaluated as important in research related to COPD are included [8].

Metabolomics studies of the lungs in COPD are still insufficient; most previous studies have focused on the metabolism of unsaturated fatty acids. Therefore, a systematic metabolomics study is needed to determine the overall metabolic flow of COPD. Using multivariate analysis and a hierarchical clustering heatmap of the lung, metabolite changes of COPD1 and COPD2 groups were found to be almost similar (Appendix A). This suggests that the metabolic changes caused by COPD in the lungs are uniquely maintained even if the lesions worsen. Changes in the fatty acid metabolism of COPD may be related to energy limitation and anoxia [22]; a condition similar to emphysema has been reported in the lungs of rats with diet restrictions [23]. Polyunsaturated fatty acids (PUFAs) have also been reported to play an important role in COPD and various other inflammatory diseases [24]. In this study, the levels of arachidonic acid, docosapentaenoic acid, and eicosapentaenoic acid were significantly increased in both COPD1 and COPD2 groups; “sphingolipid metabolism”, “fatty acid biosynthesis”, “biosynthesis of unsaturated fatty acids”, “arachidonic acid metabolism” were evaluated as important in metabolic pathway analysis. These results agree with previous reports [25,26]. Metabolomic studies on the amino acids and organic acids (including the TCA cycle) in the lungs of patients with COPD are still insufficient. The levels of almost all the investigated amino acids were elevated in patients with COPD. It has been reported that elevated phenylalanine levels are associated with inflammation [27] and are associated with COPD severity [28]. In the MSEA or metabolic pathway analysis, “phenylalanine, tyrosine, and tryptophan biosynthesis” and “phenylalanine metabolism” were evaluated as important in this study as well as in previous reports [25,26]. Interpreting our results based on reports that the administration of GABA improves lung damage [29], the increased levels of GABA may have been maintained by a defense mechanism against lung damage. Because post-translationally modified amino acids such as 3-methylhistidine are not used for de novo synthesis, they are likely to be used as biomarkers. Glutamic acid and glutamine levels are known to be related to the level of glutathione in antioxidant stress conditions [30]; the lungs may release glutamine under stress conditions [31]. In this study, the levels of glutamic acid and glutamine, GSH, and GSSG were significantly increased in the COPD group, similar to the results of previous reports [26]. The metabolic pathway related to energy metabolisms, such as the glycolysis and TCA cycle, was perturbed during COPD [32]; improvement of symptoms in COPD has been reported to be related to nutritional status [33]. In the present study, TCA-related metabolites in the lung were decreased further in the COPD2 group than the COPD1 group (i.e., as COPD worsens), but succinic acid and isocitric acid maintained an increased level (Figure 5a,b). These results due to the exacerbation of COPD suggested that metabolic disorders in the TCA cycle may be progressed and that some metabolic pathways may be deregulated.

### 3.3. Metabolic Changes in Plasma

Metabolomics studies of COPD in the blood, plasma, and plasma of patients and animal models have continued to accumulate in a variety of ways, but no biomarkers have yet been reported for COPD. Therefore, further studies must be conducted based on accumulated data. The discovery of COPD biomarkers can not only improve the lives of patients but can also help prevent COPD by detecting irreversible lung damage early on. Unlike in the lungs, changes in plasma metabolites were more sensitive to differences between the one-week (VC1 and COPD1) and two-week (VC2 and COPD2) groups than to the differences in COPD induction (Figure 5c). The lists of altered metabolites in the COPD1 and COPD2 groups were different. Previous studies on fatty acid metabolism in COPD have focused on eicosanoid, sphingolipids, and phosphatidyl derivatives; research on free fatty acids has been relatively insufficient. In this study, increased levels of docosapentaenoic acid (ω-3) and decreased levels of γ-linolenic acid (ω-6) were observed in COPD1. The other free fatty acids showed no significant changes in the COPD groups; the overall fatty acid levels tended to decrease in COPD2 rather than in COPD1. The level of docosahexaenoic acid was increased in rats treated with tobacco smoke [34], while smoking decreased the levels of PUFAs, including docosahexaenoic acid (ω-3); eicosapentaenoic acid (ω-3) [35,36] increased the levels of MUFAs, including palmitoleic acid and oleic acid [37]. The results of these studies are difficult to interpret depending on the experimental design; however, it is clear that fatty acid metabolism, including PUFAs, is related to COPD. The levels of most amino acid, including alanine, proline, serine, glycine, threonine, creatine, leucine, isoleucine, valine, glutamic acid, and glutamine, have been reported to be increased in COPD patient studies [38,39] but decreased in a COPD rat model [40]. In this study, in the COPD-induced mouse model, the levels of amino acids were decreased, or no changes were observed. The levels of most amino acids, including alanine, glycine, and serine, decreased in the COPD2 groups, whereas they were unchanged or decreased in the COPD1 group. In this study, the level of phenylalanine tended to increase more in the COPD2 group than in the COPD1 group. It has been reported that the level of phenylalanine in serum generally decreases in COPD patients, but increases significantly in GOLD stage IV patients [28,31]. Our results may support previous research showing that levels of phenylalanine increase with progression to malignant COPD. In this study, the levels of 3-methylhistidine and histidine were increased in the COPD1 group; 3-methylhistidine was also increased in the COPD2 group. Similar to our results, the level of 3-methylhistidine has been reported to be increased in the serum of patients at various stages of COPD [28]. ROC curve analysis is a statistically effective method for biomarker discovery; if the AUC is 0.7, it is acceptable for clinical application [41]. The combination of metabolites that were assessed in previous studies and in this study made it possible to differentiate COPD. Biomarker combination based on 3-methylhistidine or PUFAs showed high potential in distinguishing VC1 and COPD1 groups. On the other hand, biomarker combination based on phenylalanine showed high potential in distinguishing the COPD2 group. Specifically, α-aminobutyric acid/histidine was evaluated as a biomarker that could simultaneously distinguish COPD1 (AUC = 0.97) and COPD2 (AUC = 0.98). In the lungs of both abovementioned COPD groups, the levels of phenylalanine, 3-methylhistidine, α-linolenic acid (ω-3), DHA (ω-3), DPA (ω-3), EPA (ω-3), and arachidonic acid (ω-6) were increased. Therefore, our research team suggests that the combination of intestinal fluid metabolites consisting of 3-methylhistidine, phenylalanine, and PUFA may be a biomarker that reflects metabolic changes in the lungs. α-Aminobutyric acid/histidine is a combination of metabolites that can distinguish both the COPD1 and COPD2 groups and is important because the distinction was possible regardless of COPD exacerbation. However, this is the first study to report such findings; further research is needed.

### 3.4. Metabolic Changes in Urine

Urine is useful as a sample for disease diagnosis because it can be collected easily and conveniently in a non-invasive manner. However, most studies compared asthma with COPD, and were without a control group [42]; some research teams have conducted studies focused on COPD, but information on the urinary metabolome in COPD is still lacking. In this study, α-aminoadipic acid, glutamine, hippuric acid, homocysteine, homoserine, and *N*-methyl-D-aspartic acid were significantly different between both COPD1 and COPD2 groups. However, all their levels were increased in COPD1 and decreased in COPD2. In addition, the levels of 26 metabolites (in Appendix A), which were significantly increased in the lungs, were increased in the urine of COPD1 group, but decreased in that of the COPD2 group (Figure 5c).

Previous reports in COPD patients observed that the pyruvic acid and α-ketoglutaric acid levels [43] and histidine levels were increased [44]. In this study, the level of pyruvic acid was increased in COPD1 and COPD2 groups, but the level of α-ketoglutaric acid was increased only in the COPD1 group. Increased histidine in the lungs and urine of COPD1 may support previous reports that described an increase in histidine in asthma–COPD overlap patients along with the Th2 inflammatory response [45]. The previous findings that methionine metabolism was enhanced in COPD patients may support the increased level of methionine in this study [44]. The reports discussed so far appear to be similar to results observed with the COPD1 group in this study. The results observed with the COPD2 group were opposed to those of the COPD1 group, but the reason for this is not understood. In ROC curve analysis, candidate biomarkers (malic acid/linoleic acid (AUC = 0.96), linoleic acid/arachidonic acid (AUC = 0.95), fumaric acid/linoleic acid (AUC = 0.93), and 4-hydroxyphenyllactic acid/arachidonic acid (AUC = 0.91) in the COPD1 group, and docosahexaenoic acid/proline (AUC = 0.90) and phenylalanine/proline (AUC = 0.90) in the COPD2 group) affected by lung and plasma were found, but no metabolite combination was found that identifies both COPD1 and COPD2 groups. We do not yet know the exact reason why urinary metabolites show opposing results in COPD1 and COPD2 groups. Further studies are being planned to determine the reasons for this.

### 3.5. Integrated Metabolic Changes in Lung, Plasma, and Urine

Heat map analyses were performed to comprehensively evaluate metabolic changes using 48 metabolites determined in the lung, plasma, and urine. The levels of each metabolite were transformed into Log2FC according to the average of the COPD group compared to the average of the VC group, and Log2FC values are indicated by the numbers in the box in Figure 5c. Metabolites of the lung, plasma, and urine were hierarchically clustered into two large classes (class with an irregular pattern at the top of Figure 5c and class with a regular pattern at the bottom of Figure 5c). More than half of the 48 metabolites in the lung were increased in both COPD1 and COPD2 groups (Figure 5c). In the class with a regular pattern, the levels of 26 metabolites (histidine, 3-methylhistidine, 1-methylhistidine, valine, isoleucine, leucine, glutamic acid, glutamine, phenylalanine, tyrosine, tryptophan, glutathione, cysteine, serine, methionine, aspartic acid, asparagine, threonine, alanine, proline, 4-hydroxyproline, citrulline, ornithine, α-aminoadipic acid, creatine, and lysine) in the lung tended to increase in both COPD1 and COPD2 groups. The levels of the abovementioned 26 metabolites showed a decreasing trend in plasma, compared to that in the lung, in both the COPD1 and COPD2 groups. However, urinary levels showed completely opposite trends between the COPD1 and COPD2 groups. The levels of 26 metabolites in the urine were increased in the COPD1 group and decreased in the COPD2 group (Figure 5c). It was speculated that there was a complete reversal of metabolite levels in urine as a result of clear metabolic reprogramming between acute and exacerbated COPD. These observations make metabolic mechanisms difficult to interpret; therefore, care must be taken when conducting a biomarker discovery study.

When interpreting the pattern of metabolites in the lung metabolic maps centered on the TCA cycle, it appeared that the altered pattern of metabolites increased largely in the COPD1 group, and then somewhat relaxed in the COPD2 group (Figure 5a,b). Our research team hypothesized that metabolism was largely reflected in emphysema and inflammatory responses by COPD induction, but metabolic changes were alleviated due to adaptation and metabolic reprogramming as the state of emphysema persisted and worsened. In addition, lung metabolism in the TCA cycle was gradually downregulated from the COPD1 group to the COPD2 group, but levels of succinic acid and isocitric acid were further upregulated (Figure 5a,b). We believe that some metabolic pathways had dysfunction, or there may be a reason for their levels to remain elevated. The levels of succinic acid, isocitrate, and pyruvic acid tended to have the same pattern of increase in the lung, decrease in the plasma, and increase in the urine, in both the COPD1 and COPD2 groups. Assuming that their elevated levels in the lungs were reflected in the urine, they may be an important key in the search for common biomarkers in acute and exacerbated COPD. In ROC curve analysis, potential biomarkers based on 3-methylhistidine, phenylalanine, and PUFAs that are supported by evidence from previous studies and this study, were prominent in the plasma of COPD patients. Since previous reports on metabolite changes in urine COPD are relatively insufficient, biomarker discovery was performed based on the results of this study. As a result of performing univariate ROC curve analysis focusing on succinic acid, isocitric acid, and pyruvic acid as discussed above, succinic acid/malic acid (AUC = 1.00, Log2FC = −0.29) and succinic acid/fumaric acid (AUC = 0.98, Log2FC = −0.21) showed great capacity in discriminating between COPD1 and VC1. In addition, isocitric acid/glutamic acid (AUC = 0.92, Log2FC = 0.26), isocitric acid/arginine (AUC = 0.92, Log2FC = 0.25), isocitric acid (AUC = 0.83, Log2FC = 0.47), and pyruvic acid (AUC = 0.73, Log2FC = 0.17) showed a great capacity for discriminating between COPD2 and VC2.

## 4. Materials and Methods

### 4.1. Chemicals and Reagents

Standards of all metabolites were purchased from several vendors, such as Sigma-Aldrich (St. Louis, MO, USA) and Tokyo Chemical Industry (Kita-ku, Tokyo, Japan). HPLC-grade water, methanol (MeOH), and acetonitrile (ACN) were purchased from J.T. Baker Inc. (Phillipsburg, NJ, USA). LC-MS-grade formic acid was purchased from Thermo Scientific (Waltham, MA, USA). Spin-X centrifuge filter 0.45 μm with cellulose acetate was purchased from Costar (Corning, NY, USA). Triethylamine (TEA), hematoxylin and eosin (H&E), PPE, and LPS (*Escherichia coli* O111:B4) were purchased from Sigma-Aldrich (St. Louis, MO, USA). Phosphate-buffered saline (PBS) was purchased from Gibco (Grand Island, NY, USA). Toluene, diethyl ether (DEE), ethyl acetate (EA), dichloromethane (DCM), and sodium chloride (pesticide grade) were purchased from Kanto Chemical (Chuo-ku, Tokyo, Japan). Sodium hydroxide (NaOH) and sulfuric acid were obtained from Daejung Reagents Chemicals (Siheung, South Korea). *O*-methoxyamine hydrochloride (MO) and *N*-methyl-*N*-(*tert*-butyldimethylsilyl) trifluoroacetamide (MTBSTFA) + 1% tert-butyldimethylchlorosilane (TBDMCS) were obtained from Thermo Scientific (Bellefonte, PA, USA). All other chemicals were of analytical grade.

### 4.2. Mouse Model

Specific-pathogen-free male seven-week-old BALB/c mice were purchased from Orient Bio (Seongnam, Korea) and used after one week of acclimatization. The animal rooms were maintained at a relative humidity of 50 ± 20%, a temperature of 23 ± 3 °C, a 12 h light/dark cycle with 10 to 20 air changes/h. The animals were fed commercial rodent chow (PMI Nutritional International Inc., Richmond, IN, USA) and sterilized tap water ad libitum. All animal studies were approved by the Institutional Animal Care and Use Committee of the Korea Institute of Toxicology (Jeongeup, Republic of Korea, IACUC #1904-0143). Healthy male mice were randomly assigned to the following four experimental groups (20 mice/group): (1) vehicle control 1 (VC1); (2) once-administration group of PPE and LPS in a week (COPD1); (3) vehicle control 2 (VC2); and (4) two-dose PEE and LPS (doses minimum of one week apart) (COPD2). To establish the COPD mouse model, mice in the COPD1 and COPD2 groups were treated with PPE 0.15 unit/head via intratracheal instillation on day 1 and/or day 8, and then intratracheally treated with LPS 0.1 mg/kg on day 5 and/or day 12, using an automatic video instillator [46,47]. The VC1 and VC2 groups were instilled with an equal amount of saline instead of PPE and LPS via the same route. The experimental animals were necropsied on day 8 (VC1 and COPD1 groups) and day 15 (VC2 and COPD2 groups). The experimental schemes of the mouse model are shown in Figure 6.

### 4.3. Sample Collection

From each group, 10 animals were used for the sample collection. The urine sample was collected from an individual mouse prior to necropsy according to standard procedure [48]. Fresh urine samples were clarified by centrifugation at 3000 rpm for 5 min at 4 °C. After urine collection, the mice were anesthetized with an overdose of isoflurane (Hana Pharm Co., Ltd., Hwa-Sung, Korea), and blood samples (approximately 0.5 mL) were collected from the posterior vena cava of mice using a 26-gauge syringe from each mouse. The whole blood samples were collected into complete blood count bottles containing EDTA-2K (BD, Franklin Lakes, NJ, USA) and centrifuged at 3000 rpm for 10 min at 4 °C to separate the plasma. The lung was removed and then weighed immediately after collection of full blood samples. The urine, plasma, and lung samples were frozen in liquid nitrogen and stored at −80 °C until analysis.

### 4.4. Lung Function Measurement

The 10 animals (except for sample collection) of each group were used to measure lung function. The lung function was investigated by using an eSpira Forced Manoeuvres System (EMMS, Hants, UK) following the manufacturer’s protocol. Briefly, the mice were anesthetized by intraperitoneal injection of 24 mg/kg of alfaxalone (Alfaxan; Careside, Gyeonggi-do, Korea) and 14 mg/kg of xylazine (Rompun; Bayer Korea, Seoul, Korea). Tracheostomized mice were intubated and placed in the body plethysmograph. The average breathing frequency was set to 150 breaths/min. Forced vital capacity (FVC), total lung capacity (TLC), and forced expiratory volume in first 100 milliseconds of exhalation (FEV100) were recorded during fast flow volume maneuver. Functional residual capacity (FRC) was measured from the Boyle’s law.

### 4.5. BALF Collection and Cell Counting

After measurement of lung function, BALF was collected with 0.7 mL PBS for three times in each right lung of the mouse. Total cells in BALF were quantified and measured using a NucleoCounter (NC-250; Chemometec, Gydevang, Denmark). The cells in BALF were then subjected to Diff-Quik staining (Sysmex, Kobe, Japan) for differential counting under microscope (BX51; Olympus, Tokyo, Japan). At least 200 cells/sample including macrophage, neutrophil, lymphocyte, and eosinophil were scored.

### 4.6. Histological Examination

After the left lung (which had not been lavaged) was removed, it was fixed in 10% neutral-buffered formalin and embedded in paraffin. Paraffin-embedded tissue blocks were sectioned into 4-μm-thick slices. Sections from each mouse were stained with hematoxylin and eosin for histological examination. Stained lung tissue slides were evaluated under a light microscope at 200× magnification. The MLI is a common method used to quantify lung injury in histopathological images. The number of intersections between the line of the alveolar walls and the grid was quantified in 10 randomly chosen fields of view at 100× magnification per animal, as described previously [49,50].

### 4.7. Sample Preparation of Lung, Plasma, and Urine

Lungs were homogenized in HPLC-grade water using an ultrasonicator (IKA-Werke GmbH & Co. KG, Staufen, Germany). Lung extract, serum, and urine were transferred to the Spin-X centrifuge filter, centrifuged at 12,300× *g* for 10 min. For deproteinization of lung extract and plasma, three volumes of cold ACN were added to the filtered lung extract and plasma and vortex-mix for 1 min. In the profiling analysis of 24 fatty acids and 18 organic acids using GC-MS/MS, derivatization of fatty acid and organic acid were performed through a methoxycarbonylation and silylation reaction using MO and MTBSTFA. The extract of lung, serum, or urine with pentadecanoic acid (0.1 µg), 3,4-methoxybenzoic acid (0.1 µg), and ^13^C_2_-succinic acid (1.0 µg) as internal standard was spiked into HPLC-grade water (1 mL). The aqueous phase was basified to pH > 12 using 5 M NaOH solution; methoxyamine hydrochloride 100 μL (10 mg/mL) was added and the mixture was heated at 60 °C for 60 min. The aqueous phase was acidified to pH < 2 using 10% sulfuric acid solution; three liquid–liquid extractions (LLE) were performed sequentially by DEE (3 mL), EA (2 mL), and DEE: DCM (2:1, *v*/*v*) (3 mL). Pooled extracts of LLE were evaporated under a gentle stream of nitrogen (N_2_) at 40 °C to dryness. Toluene (20 μL) and MTBSTFA (20 μL) were added to the residue; the mixture was heated at 60 °C for 60 min. In the profiling analysis of other metabolites, including amino acids, using LC-MS/MS, the extract of lung, serum, or urine with ^13^C_1_-phenylalanine (25 ng) as IS was spiked into HPLC-grade water (1 mL). LC-MS/MS analysis was performed after filtration and deproteinization of samples.

### 4.8. GC-MS/MS

The profiling analyses for fatty acids and organic acids were performed with GCMS-TQ8040 (Shimadzu Corp., Kyoto, Japan) interfaced with a triple quadrupole mass spectrometer in electron impact mode at 70 eV. An Ultra-2 (5% phenyl–95% methylpolysiloxane bonded phase; 25 m × 0.20 mm i.d., 0.11-μm film thickness) cross-linked capillary column (Agilent Technologies, Palo Alto, CA, USA) was used for chromatographic separation analysis. The carrier gas (helium), in which the linear velocity flow control mode was applied was used at a flow rate of 0.5 mL/min. The split-injection mode was performed at a split ratio of 10:1. The injector, interface, and ion source temperatures were maintained at 260 °C, 300 °C, and 230 °C, respectively. The oven temperature for fatty acid profiling analysis started at 100 °C (held for 3 min), increased to 260 °C (1.5 °C/min), increased to 300 °C (20 °C/min), and was then held for 5 min. The oven temperature of the organic acid profiling analysis started at 100 °C (held for 2 min), increased to 250 °C (10 °C/min), increased to 300 °C (20 °C/min), and was then held for 3 min. The scanned mass range was 50–750 u at a rate of 15,000 µ/s. In SRM modes, one quantitative ion for each metabolite was used for peak quantification.

### 4.9. LC-MS/MS

LC-MS/MS analysis was performed using a Shimadzu Nexera UPLC system (Shimadzu Corp., Kyoto, Japan) with an LCMS-8050 triple quadrupole mass spectrometer (Shimadzu Corp., Kyoto, Japan). Chromatographic separation was achieved using an Intrada amino acid column (150 mm × 2 mm, 3 µm). Nebulizing and drying gas flow was 3.0 L/min and 10.0 L/min, respectively. The pressure of the collision-induced dissociation (CID) gas was 270 kPa. The interface, desolvation line (DL) and heat block temperature were set at 200 °C, 200 °C, and 300 °C, respectively. Mobile phase A was composed of ACN/tetrahydrofuran/25 mM ammonium formate/formic acid (9/75/16/0.3, *v*/*v*/*v*/*v*) and mobile phase B was composed of ACN/100 mM ammonium formate (20/80, *v*/*v*). The mobile phase gradient program, initially set at 0% of mobile phase B, increased to 17% of mobile phase B (2.5–6.5 min), increased to 100% of mobile phase B (6.5–10 min), held for 5 min, and decreased to the initial 0% of mobile phase B, followed by a 3 min re-equilibration period. The total flow rate was set at 0.6 mL/min; the column temperature was maintained at 35 °C. In SRM modes with electrospray ionization (ESI) mode, one quantitative ion for each metabolite was used for peak quantification.

### 4.10. Statistical Analyses

Data from animal experiments are presented as mean ± standard deviation (SD); statistical comparisons were performed using one-way analysis of variance (ANOVA) followed by a Student’s *t*-test. The comparison results are indicated as *p* < 0.05, or *p* < 0.01. Statistical analyses of metabolomic data, including Wilcoxon rank-sum tests, principal component analysis (PCA), partial least squares-discriminant analysis (PLS-DA), hierarchical clustering heatmaps, and univariate and multivariate receiver operating characteristic (ROC) curve analyses were performed using MetaboAnalyst 4.0 (https://www.metaboanalyst.ca, accessed on 27 January 2022) [51].

## 5. Conclusions

This study was a metabolomic study on the lungs, plasma, and urine in a PPE/LPS-induced COPD exacerbation model. In addition, the COPD acute model (COPD1 group) and COPD acute exacerbation model (COPD2 group) were evaluated to investigate the differences in COPD severity. A study, designed as described above, to discover potential biomarkers in COPD models has not yet been conducted. Metabolite analysis revealed that metabolic changes in the lungs showed similar patterns in both COPD1 and COPD2 groups. Metabolic changes between the COPD1 and COPD2 groups showed large differences in plasma and showed greater differences in urine. The levels of more than half of the urinary metabolites investigated showed an opposite pattern between the COPD1 and COPD2 groups. The levels of metabolites of COPD1 and COPD2 were similar in the lungs; however, the exact reason for this large difference in patterns of metabolite levels between the lung, plasma, and urine has not yet been determined. Our research team speculated that metabolic reprogramming occurred between the COPD1 and COPD2 groups, and that the lungs maintained homeostasis for emphysema and inflammatory responses, which did not happen in plasma and urine. Compared to previous reports, several types of plasma biomarkers, including phenylalanine, 3-methylhistidine, and PUFAs, have been proposed in COPD1 or COPD2. α-Aminobutyric acid/histidine (AUC > 0.97) was first discovered as a serous potential biomarker that identifies both COPD1 and COPD2. In addition, metabolite combinations of succinic acid, isocitric acid, and pyruvic acid have been proposed as urinary biomarkers that reflect lung metabolism in COPD1 or COPD2. Therefore, this study suggests that plasma and urine biomarkers that are directly related to lung metabolism provide new insights for the diagnosis, treatment, and severity evaluation of COPD.

## Figures and Tables

**Figure 1 ijms-23-02748-f001:**
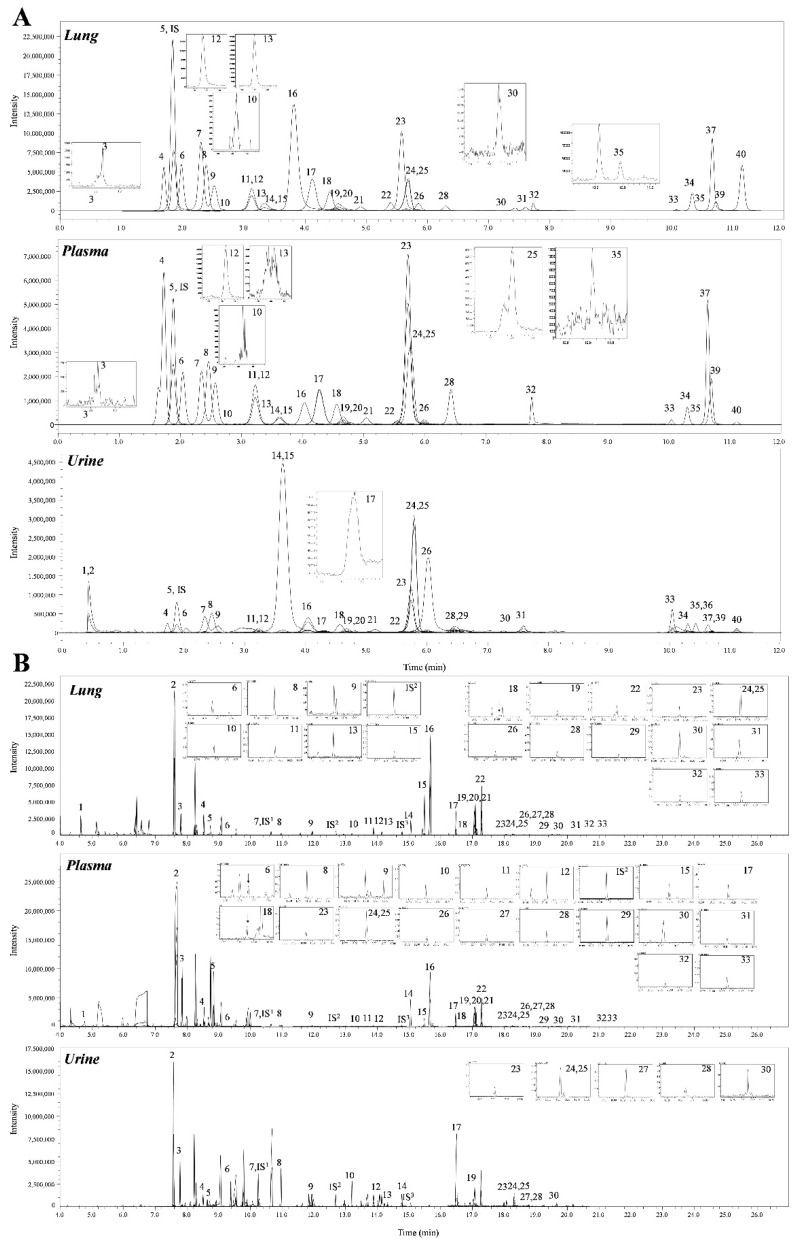
Representative SRM chromatograms of LC-MS/MS (**A**) and GC-MS/MS as MO/TBDMS derivatives (**B**) in lung, plasma, and urine. The peak numbers of LC-MS/MS correspond to those in Appendix A, the peak numbers of GC-MS/MS correspond to those in Appendix A.

**Figure 2 ijms-23-02748-f002:**
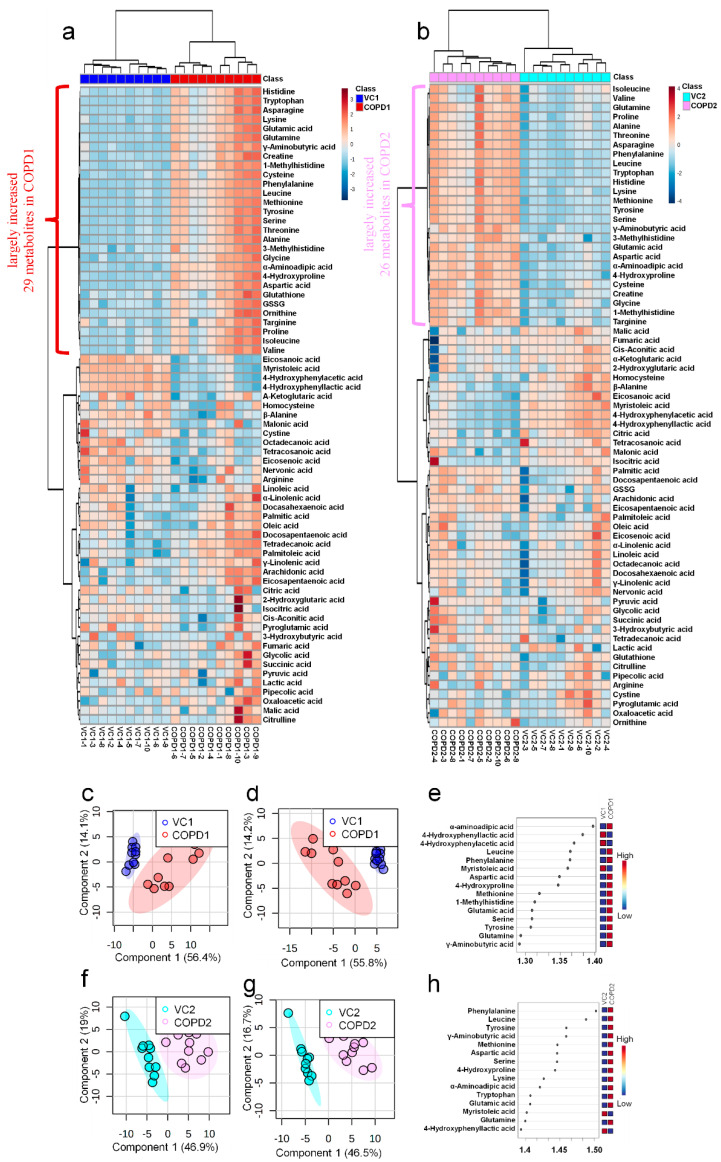
Hierarchical clustering heatmap (**a**), PCA scores plot (**c**), PLS-DA scores plot (**d**), and VIP score of top-15 metabolites in PLS-DA (**e**) of lung in COPD1 and VC1 groups. Hierarchical clustering heatmap (**b**), PCA scores plot (**f**), PLS-DA scores plot (**g**), and VIP score of top-15 metabolites in PLS-DA (**h**) of lung in COPD2 and VC2 groups.

**Figure 3 ijms-23-02748-f003:**
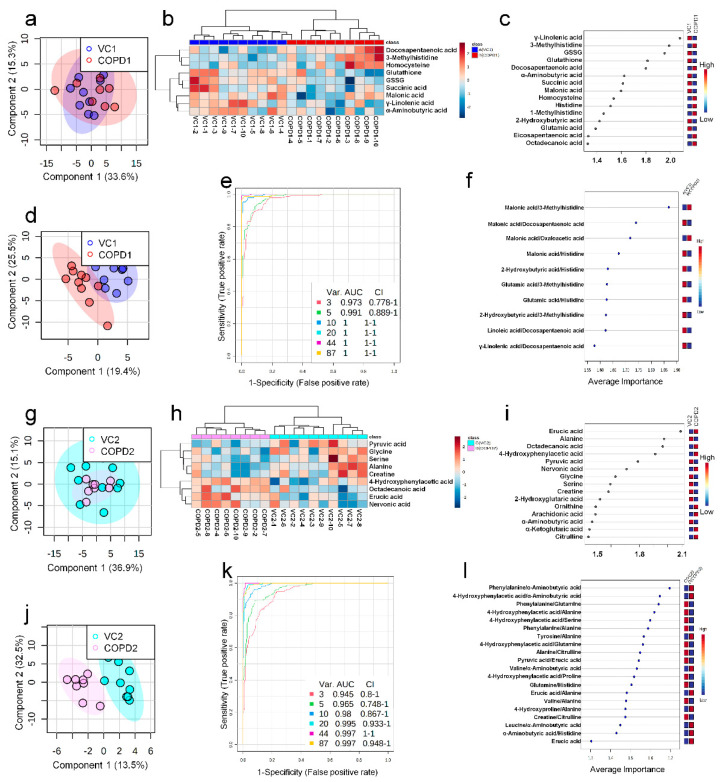
PCA scores plot (**a**), hierarchical clustering heatmap of 9 significant metabolites (**b**), VIP score of top-15 metabolites in PLS-DA (**c**), PLS-DA scores plot (**d**), multivariate ROC curve analysis (**e**), average importance of 10 metabolite combination in multivariate ROC curve analysis (**f**) of plasma in COPD1 and VC1 groups. PCA scores plot (**g**), hierarchical clustering heatmap of 9 significant metabolites (**h**), VIP score of top-15 metabolites in PLS-DA (**i**), PLS-DA scores plot (**j**), multivariate ROC curve analysis (**k**), average importance of 20 metabolite combinations in multivariate ROC curve analysis (**l**) of plasma in COPD2 and VC2 groups.

**Figure 4 ijms-23-02748-f004:**
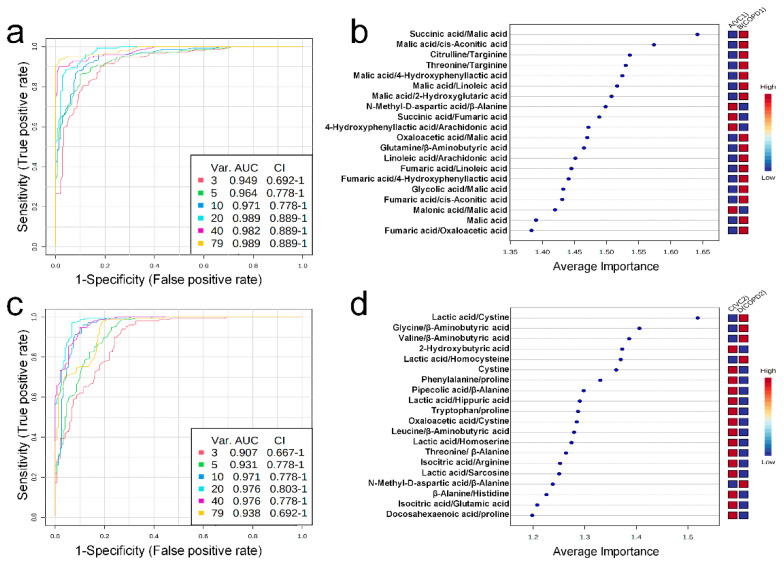
Multivariate ROC curve analysis (**a**), average importance of metabolite combinations of 20 types in multivariate ROC curve analysis (**b**) of urine in COPD1 and VC1 groups. Multivariate ROC curve analysis (**c**), average importance of metabolite combinations of 20 types in multivariate ROC curve analysis (**d**) of urine in COPD2 and VC2 groups.

**Figure 5 ijms-23-02748-f005:**
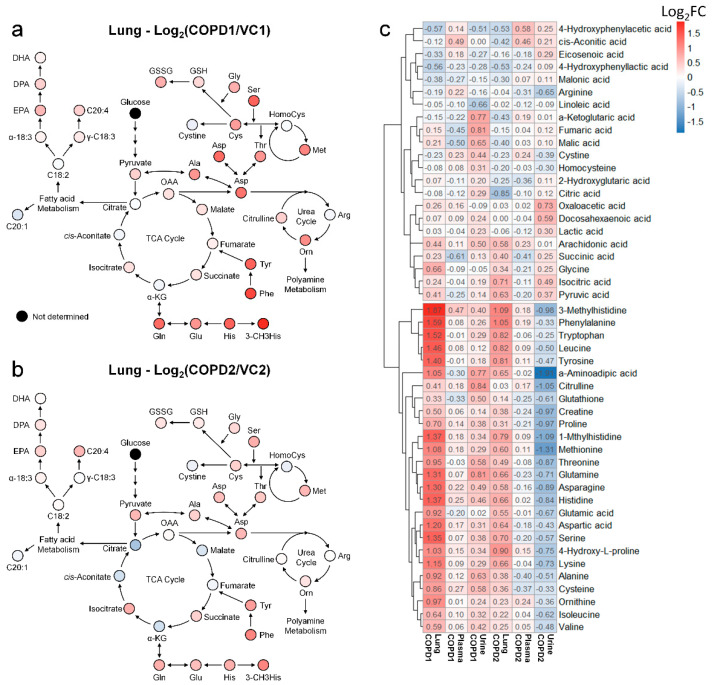
Metabolic pathway maps of lung in COPD1 (**a**) and COPD2 (**b**) groups. Hierarchical clustering heatmap of lung, plasma, and urine in COPD1 and COPD2 groups (**c**). An indicator expressing the value of Log2FC in color (red; increase and blue; decrease) was used in the comparison of COPD1 and COPD2 with VC1 and VC2, respectively.

**Figure 6 ijms-23-02748-f006:**
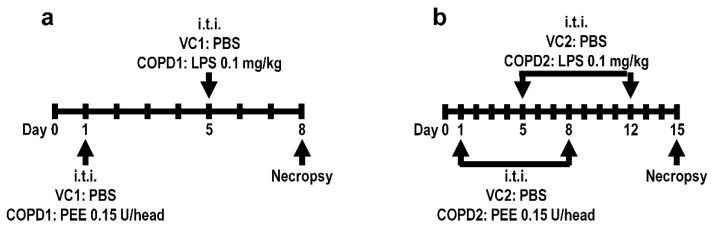
Mouse model experimental schemes. Vehicle control (VC1) and once administration group of PPE and LPS in a week (COPD1) within a week (**a**), and vehicle control (VC2) and two-dose PEE and LPS (doses minimum of one week apart) (COPD2) within two weeks (**b**).

## Data Availability

Not applicable.

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
