# Peer review of "Comprehensive Targeted Metabolomic Study in the Lung, Plasma, and Urine of PPE/LPS-Induced COPD Mice Model"

_ijms, 2022, doi:10.3390/ijms23052748_

Round 1

Reviewer 1 Report

The manuscript by Kim et al. presents a target metabolomics study based on GC-MS and LC-MS methods implemented into the field of chronic obstructive pulmonary disease. Authors used metabolomics approaches for investigation of new putative biomarkers of this disease in various biological samples – lung tissue, serum, urine, obtained from the PPE/LPS-induced COPD mice model. The manuscript is interesting, attractive for the readers and fits with with the journal (International Journal of Molecular Sciences). However, the quality is not enough to justify publication in the present form. I recommend major revision. Find below some specific suggestions to improve the quality of the paper.

Introduction:

The information about the number of patients suffering from COPD should be added into the manuscript, and/or prognosis for the future should be added too.

A brief discussion about the methods used in metabolomics – i.e. MS and NMR should be added into the Introduction.

The papers dealing with metabolomics studies of COPD should be cited in the Introduction.

Studies dealing with analysis of metabolites in various biological samples should be discussed.

Page 2 – line 61:  performed´, - please correct this error

Results:

Page 2 – line 94 – the abbreviation MLI should be explained

Page 6 – line 190 – 191 – How was performed the analysis of creatinine? The information is missing.

Discussion:

The benefits of the mice model used for the investigation should be discussed more in details.

I was wondering, why the authors have not investigated the analytes from kynurenine pathway?

Materials and Methods:

Page 12 – line 414 – DW – I do not think that this abbreviation is correct. I recommend to use only the term HPLC-grade water in the manuscript.

4.2. Mouse model:

Why were only seven-week old mice used?

4.3 Sample collection:

Page 13 – line 456 – How was the blood collected?

The information about collection of the bronchoalveolar lavage fluid is missing.

4.5 Sample preparation of lung, serum, and urine

The preparation of the lung extract is missing.

The information about preparation of the biological samples is missing. How were the samples (serum, urine, lung extract) treated before the analysis?

Page 13 – line 478 - …. as previously reported by our research team. – Appropriate reference is missing.

Page 13 – line 481 - …. as previously reported by our research team. – Appropriate reference is missing.

The conditions for the derivatization procedure are missing.

4.6 GC-MS/MS and 4.7. LC-MS/MS

The information about the used SRM ion transitions of the significantly changed metabolites should be added into the Supplementary material.

Illustrative GC-MS and LC-MS chromatograms should be added into the manuscript.

Author Response

Response to Reviewer 1 Comments

Introduction:

Point 1: The information about the number of patients suffering from COPD should be added into the manuscript, and/or prognosis for the future should be added too.

Response 1:

As, your comments, the following sentences and references have been added in the Introduction and References: The prevalence and incidence of COPD in the worldwide is approximately 174 million people [4]. Also, the annual death from COPD in the United States is more than 120,000 people and about 3.2 million people globally [4,5].

[4] Soriano, J.B. An Epidemiological Overview of Chronic Obstructive Pulmonary Disease: What Can Real-Life Data Tell Us about Disease Management? COPD 2017, 14, S3-S7.

[5] Pauwels, R.A.; Buist, A.S.; Calverley, P.M.; Jenkins, C.R.; Hurd, S.S.; GOLD Scientific Committee. Global strategy for the diagnosis, management, and prevention of chronic obstructive pulmonary disease. NHLBI/WHO Global Initiative for Chronic Obstructive Lung Disease (GOLD) Workshop summary. Am. J. Respir. Crit. Care Med. 2001, 163, 1256-1276.

Point 2: A brief discussion about the methods used in metabolomics – i.e. MS and NMR should be added into the Introduction.

Response 2: As, your comments, we revised and added a sentences about methods used in metabolomics in "Introduction".

In metabolomics, profile analysis is generally performed using GC and LC-based mass spectrometry or NMR [12]. It is important to select an analytical instrument based on the properties of the metabolite. In general, LC-based analysis is preferred for non- volatile and polar metabolites [12]. For some metabolites, such as fatty acids and organic acids, GC-based analysis after derivatization is preferred because it is generally a more sensitive and selective with the improved MS properity than those of non-derivatization. [13].

[12] Obata, T.; Fernie, A.R. The use of metabolomics to dissect plant responses to abiotic stresses. Cell. Mol. Life Sci. 2012, 69, 3225-3243.

[13] Fernie, A.R.; Trethewey, R.N.; Krotzky, A.J.; Willmitzer, L. Metabolite profiling: from diagnostics to systems biology. Nat. Rev. Mol 2004, 5, 763-769.

Point 3: The papers dealing with metabolomics studies of COPD should be cited in the Introduction. Studies dealing with analysis of metabolites in various biological samples should be discussed.

Response 3: As, your comments, we revised and added a sentences about the analysis of metabolites in various biological samples in "Introduction".

Many metabolomics-based COPD studies have been performed in serum, plasma, lung, and exhaled breath condensate, but the biochemical mechanisms of COPD are not yet fully understood [8].

Point 4: Page 2 – line 61: performed´, - please correct this error

Response 4: As your comments, we revised.

Results:

Point 5: Page 2 – line 94 – the abbreviation MLI should be explained

Response 5: As, your comments, it has been corrected as pointed out in the ‘2.4 Histological examination’.

These histological changes were reflected in an increase in the mean linear intercept (MLI, Figure S4 e).

Point 6: Page 6 – line 190 – 191 – How was performed the analysis of creatinine? The information is missing.

Response 6: Quantitative analysis of creatinine was also performed when performing profiling analysis by LC-MS.

Discussion:

Point 7: The benefits of the mice model used for the investigation should be discussed more in details.

Response 7: As, your comments, the following sentence has been added in the ‘3.1.

PPE/LPS-induced COPD exacerbation model’

Also, severe exacerbation of COPD patients required long term exposure to CS, air pollution, fine dust, and many chemicals, whereas our COPD mouse model can on some extent reflects the pathology and physiology of human COPD in a short modeling time.

Point 8: I was wondering, why the authors have not investigated the analytes from kynurenine pathway?

Response 8: We agree to your comment for importance of kynurenine pathway. Unfortunately, in this study, we have only performed for altered energy metabolism as target metabolomics analysis in COPD. Although metabolites in kynurenine pathway was not performed in this study, our research team is preparing the next study to fill the gaps in the current study.

Materials and Methods:

Point 9: Page 12 – line 414 – DW – I do not think that this abbreviation is correct. I recommend to use only the term HPLC-grade water in the manuscript.

Response 9: As your comments, we revised.

4.2. Mouse model:

Point 10: Why were only seven-week old mice used?

Response 10: The 6~8-week old mice were commonly used for animal studies and for which there is a large historical database. The age of mice was chosen based on the previous studies for experimental emphysema model (Tanaka et al., 2014; Kim et al., 2015).

Tanaka, K.-I; Kurotsu, S.; Asano, T.; Yamakawa, N.; Kobayashi, D.; Yamashita, Y.; Yamazaki, H.; Ishihara, T.; Watanabe, H.; Maruyama, T.; Suzuki, H.; Mizushima, T. Superiority of pulmonary administration of mepenzolate bromide over other routes as treatment for chronic obstructive pulmonary disease. Sci. Rep. 2014, 4, 4510.

Kim, Y.-S.; Kim, J.-Y.; Huh, J.W.; Lee, S.W.; Choi, S.J.; Oh, Y.-M. The Therapeutic Effects of Optimal Dose of Mesenchymal Stem Cells in a Murine Model of an Elastase Induced-Emphysema. Tuberc. Respir. Dis. (Seoul) 2015, 78, 239-245.

4.3 Sample collection:

Point 11: Page 13 – line 456 – How was the blood collected?

Response 11: As, your comments, the following sentences have been added in the ‘4.3. Sample collection’

blood samples (approximately 0.5 ml) were collected from the posterior vena cava of mice using a 26-gauge syringe from each mouse. The whole blood samples were collected into complete blood count bottles containing EDTA-2K (BD, Franklin Lakes, NJ, USA) and centrifuged at 3000 rpm for 10 min at 4°C to separate the plasma.

Point 12: The information about collection of the bronchoalveolar lavage fluid is missing.

Response 12: As, your comments, the following sentences have been added in the ‘4.5. BALF collection and cell counting’

After measurement of lung function, BALF was collected with 0.7 ml PBS for three times in each right lung of the mouse. Total cell in BALF were quantified and measured using a NucleoCounter (NC-250; Chemometec, Gydevang, Denmark). The cells in BALF were then subjected to Diff-Quik staining (Sysmex, Kobe, Japan) for differential counting under microscope (BX51; Olympus, Tokyo, Japan). At least 200 cells/sample included macrophage, neutrophil, lymphocyte, and eosinophil were scored.

4.5 Sample preparation of lung, serum, and urine

Point 13: The preparation of the lung extract is missing. The information about preparation of the biological samples is missing. How were the samples (serum, urine, lung extract) treated before the analysis?

Response 13: As, your comments, we added a sentence about the preparation of the lung extract, as follows

Lungs were homogenized in HPLC-grade water using an ultrasonicator (IKA-Werke GmbH & Co.KG, Staufen, Germany). Lung extract, serum, and urine were transferred to the Spin-X centrifuge filter, centrifuged at 12,300 g for 10 min. To deproteinization of lung extract and plasma, three volumes of cold ACN were added to the filtered lung extract and plasma and vortex-mix for 1 min.

Point 14: Page 13 – line 478 - …. as previously reported by our research team. – Appropriate reference is missing. Page 13 – line 481 - …. as previously reported by our research team. – Appropriate reference is missing. The conditions for the derivatization procedure are missing.

Response: 14: As, your comments, we revised and added a sentences about The conditions for the derivatization procedure of GC-MS/MS, as follows

Lungs were homogenized in HPLC-grade water using an ultrasonicator (IKA-Werke GmbH & Co.KG, Staufen, Germany). Lung extract, serum, and urine were transferred to the Spin-X centrifuge filter, centrifuged at 12,300 g for 10 min. To deproteinization of lung extract and plasma, three volumes of cold ACN were added to the filtered lung extract and plasma and vortex-mix for 1 min. In the profiling analysis of 24 fatty acids and 18 organic acids using GC-MS/MS, derivatization of fatty acid and organic acid were performed through a methoxycarbonylation and silylation reaction using MO and MTBSTFA. The extract of lung, serum, or urine with pentadecanoic acid (0.1 µg), 3,4-methoxybenzoic acid (0.1 µg), and 13C2-succinic acid (1.0 µg) as internal standard as spiked into HPLC-grade water (1 mL). The aqueous phase is basified to pH>12 using 5 M NaOH solution, added methoxyamine hydrochloride 100 μL (10 mg/mL), heated at 60°C for 60 min. The aqueous phase is acidified to pH<2 using 10% sulfuric acid solution, three times of liquid-liquid extraction (LLE) were performed sequentially by DEE (3 mL), EA (2 mL), and DEE: DCM (2:1, v/v) (3 mL). Pooled extracts of LLE were evaporated under a gentle stream of nitrogen (N2) at 40°C to dryness. Toluene (20 μL) and MTBSTFA (20 μL) were added to the residue, and the mixture was heated at 60°C for 60 min. In profiling analysis of other metabolites, including amino acids, using LC-MS/MS, the extract of lung, serum, or urine with 13C1-phenylalanine (25 ng) as IS was spiked into HPLC-grade water (1 mL). LC-MS/MS analysis has performed after filtration and deproteinization of samples.

4.6 GC-MS/MS and 4.7. LC-MS/MS

Point 15: The information about the used SRM ion transitions of the significantly changed metabolites should be added into the Supplementary material.

Response 15: As, your comments, we revised and added a sentences about SRM ion transitions table Supplementary material.

Supplementary Table 8. Selected reaction monitoring conditions for 45 organic metabolites include amino acids and IS profiling analysis by LC-MS/MS

No

Metabolites

Electro spray ionization mode

SRM (m/z)

CE (V)

1

Hydroxyhippuric acid

+

195.90>121.15

-13

2

Hippuric acid

+

180.20>105.15

-13

3

Pyroglutamic acid

+

130.00>84.20

-13

4

Tryptophan

+

205.10>188.00

-10

5

Phenylalanine

+

166.10>120.00

-15

6

Tyrosine

+

182.10>136.10

-12

7

Leucine

+

132.00>43.20

-23

8

Methionine

+

150.00>56.10

-16

9

Isoleucine

+

132.10>69.20

-16

10

Homocysteine

+

135.80>90.10

-10

11

Valine

+

118.10>72.10

-10

12

Cysteine

+

122.00>59.10

-22

13

GSH reduced (GSH)

+

307.80>179.15

-12

14

a-Aminoadipic acid

+

161.90>98.10

-14

15

Pipecolic acid

+

130.10>84.10

-15

16

Glutamic acid

+

148.00>84.00

-15

17

Proline

+

116.10>70.00

-15

18

Threonine

+

120.00>74.20

-11

19

4-Hydroxyproline

+

132.00>86.10

-14

20

Aspartic acid

+

134.00>74.15

-13

21

Alanine

+

90.00>44.15

-11

22

Serine

+

106.00>60.15

-11

23

Glutamine

+

147.00>84.20

-16

24

Glycine

+

75.80>30.15

-12

25

Creatine

+

132.20>90.15

-13

26

Asparagine

+

133.00>74.15

-15

27

Creatinine

+

114.00>86.15

-13

28

Citrulline

+

176.00>70.20

-22

29

b-Aminoisobutyric acid

+

104.00>86.15

-10

30

b-Alanine

+

89.90>72.20

-11

31

γ-Aminobutyric acid

+

104.00>87.10

-11

32

GSH oxidized (GSSG)

+

613.00>484.10

-16

33

1-Methylhistidine

+

169.90>124.15

-14

34

Histidine

+

156.00>110.00

-15

35

3-Methylhistidine

+

169.90>95.20

-30

36

NG-Methylarginine

+

189.30>70.20

-23

37

Lysine

+

147.10>84.10

-15

38

γ-Hydroxylysine

+

162.90>128.20

-13

39

Ornithine

+

132.90>70.10

-17

40

Arginine

+

175.10>70.10

-25

41

Sarcosine

+

89.90>44.15

-13

42

N-Methyl-DL-aspartic acid

+

148.00>87.95

-10

43

Homoserine

+

120.10>74.15

-12

44

a-Aminobutyric acid

+

104.00>58.25

-11

45

5-Hydroxytryptophan

+

220.90>204.25

-10

IS

13C1-Phenylalanine (IS)

+

167.10>120.10

-15

Supplementary Table 9. Selected reaction monitoring conditions for the 24 fatty acid, 18 organic acid and 3 ISs profiling analysis by GC-MS/MS

No

Metabolites

Ionization mode

SRM (m/z)

CE (V)

1

Pyruvic acid

EI

174.00>74.10

15

2

Lactic acid

EI

261.00>147.10

15

3

Glycolic acid

EI

247.00>147.10

15

4

2-Hydroxybutyric acid

EI

275.00>147.10

15

5

3-Hydroxybutyric acid

EI

275.00>159.20

5

6

Malonic acid

EI

275.00>73.10

20

7

Succinic acid

EI

289.00>147.10

10

8

Fumaric acid

EI

287.00>147.10

15

9

Oxaloacetic acid

EI

332.00>147.10

10

10

α-Ketoglutaric acid

EI

346.00>156.10

10

11

C14:0

EI

285.00>131.10

10

12

4-Hydroxyphenylacetic acid

EI

323.00>205.10

10

13

Malic acid

EI

419.00>115.10

10

14

2-Hydroxyglutaric acid

EI

433.00>245.20

15

15

C16:1

EI

311.00>131.10

10

16

C16:0

EI

313.00>131.10

10

17

cis-Aconitic acid

EI

459.00>147.10

20

18

γ-C18:3

EI

335.00>243.20

5

19

C18:2

EI

337.00>131.20

15

20

C18:1

EI

339.00>131.10

10

21

a-C18:3

EI

335.00>75.00

30

22

C18:0

EI

341.00>131.10

10

23

4-Hydroxyphenyllactic acid

EI

467.00>439.30

5

24

Citric acid

EI

459.00>147.10

20

25

Isocitric acid

EI

459.00>147.20

20

26

C20:4

EI

361.00>269.00

5

27

C20:5

EI

359.00>75.00

30

28

C20:1

EI

367.00>131.20

10

29

C20:0

EI

369.00>131.10

15

30

C22:6

EI

385.00>75.10

24

31

C22:5

EI

387.00>75.20

20

32

C22:1

EI

395.00>131.10

15

33

C24:1

EI

423.00>131.10

15

34

Acetoacetic acid

EI

188.00>89.10

9

35

C6:0

EI

173.00>131.10

5

36

C8:0

EI

201.00>131.10

5

37

C10:0

EI

229.00>131.10

5

38

C12:0

EI

257.00>131.10

10

39

C14:1

EI

283.00>131.10

10

40

C22:0

EI

397.00>131.10

10

41

C24:0

EI

425.00>131.20

10

42

C26:0

EI

453.00>131.20

15

43

C15:0 (IS)3

EI

299.00>131.10

10

44

2-13C- Succinic acid (IS)

EI

291.00>147.10

10

45

3,4-Dimethoxybenzoic acid (IS)

EI

239.00>195.10

10

Point 16: Illustrative GC-MS and LC-MS chromatograms should be added into the manuscript.

Response 16: As, your comments, we added Figure 1 about representative SRM chromatograms of LC-MS/MS and GC-MS/MS as MO/TBDMS derivatives in lung, plasma, and urine.

Figure 1. Representative SRM chromatograms of LC-MS/MS (A) and GC-MS/MS as MO/TBDMS derivatives (B) in lung, plasma, and urine. The peak numbers of LC-MS/MS correspond to those in Supplementary Table 8, the peak numbers of GC-MS/MS correspond to those in Supplementary Table 9.

Reviewer 2 Report

The authors designed COPD mouse models and employed targeted metabolomic tools to study the metabolic mechanisms of COPD, and spent  substantial amount of effort to propose biomarkers for this disease. The design of the study is incomplete for proposing either biological mechanisms or biomarkers.

  1. Although the title says 'targeted metabolomic' study, there are no explanatory descriptions on what sets of targets were specifically selected and why so. I could only see “24 fatty acids” and “18 organic acids” were mentioned in the method section at the end of the manuscript for GC-MS analysis. For LC-MS, sets of targets were not mentioned at all. The readers can probably guess what was targeted from the presented figures and data. But the lack of description and appropriate reasoning made this paper very confusing.
  2. The proposed sets of biomarkers can be of low specificity due to the design of the study. An appropriate positive control needs to be selected ( i.e a non-COPD lung disease model) and compared in parallel with current data.
  3. MetaboAnalyst 4 was used for statistical analysis but no reference was cited.

Author Response

Response to Reviewer 2 Comments

Point 1: Although the title says 'targeted metabolomic' study, there are no explanatory descriptions on what sets of targets were specifically selected and why so. I could only see “24 fatty acids” and “18 organic acids” were mentioned in the method section at the end of the manuscript for GC-MS analysis. For LC-MS, sets of targets were not mentioned at all. The readers can probably guess what was targeted from the presented figures and data. But the lack of description and appropriate reasoning made this paper very confusing.

Response 1: As, your comments, we revised and added a sentences about descriptions on what sets of targets were specifically selected and why so in "3.2. Metabolic changes in lung"

3.2. Metabolic changes in lung

In this study, target metabolomics analysis was performed on 44 organic metabolites including amino acids using LC-MS/MS and 18 organic acids and 24 fatty acids using GC-MS/MS. A total of 86 metabolites can be evaluated importantly for their relevance to energy metabolism based on the TCA cycle, and metabolites previously evaluated as important in research related to COPD are included [8].

Point 2: The proposed sets of biomarkers can be of low specificity due to the design of the study. An appropriate positive control needs to be selected (i.e a non-COPD lung disease model) and compared in parallel with current data.

Response 2: Our research team are planning the next study to make up for the shortcomings of the current study. In the futher study, efforts will be made to select an appropriate positive control group.

Point 3: MetaboAnalyst 4 was used for statistical analysis but no reference was cited.

Response 3: As, your comments, we added a reference about MetaboAnalyst 4.0.

[51] Chong, J.; Wishart, D.S.; Xia, J. Using MetaboAnalyst 4.0 for comprehensive and integrative metabolomics data analysis. Curr. Protoc. Bioinform. 2019, 68, e86.

Round 2

Reviewer 1 Report

I would like to thank the authors for judiciously answering the questions. I have no objections to publication of the manuscript at this point.